# Infants’ First Solid Foods: Impact on Gut Microbiota Development in Two Intercontinental Cohorts

**DOI:** 10.3390/nu13082639

**Published:** 2021-07-30

**Authors:** Chiara-Maria Homann, Connor A. J. Rossel, Sara Dizzell, Liene Bervoets, Julia Simioni, Jenifer Li, Elizabeth Gunn, Michael G. Surette, Russell J. de Souza, Monique Mommers, Eileen K. Hutton, Katherine M. Morrison, John Penders, Niels van Best, Jennifer C. Stearns

**Affiliations:** 1Department of Medicine, McMaster University, Hamilton, ON L8N 3Z5, Canada; chiara.m.homann@hotmail.com (C.-M.H.); sara.dizzell@gmail.com (S.D.); surette@mcmaster.ca (M.G.S.); 2Department of Pediatrics, McMaster University, Hamilton, ON L8S 4K1, Canada; gunne@mcmaster.ca (E.G.); morriso@mcmaster.ca (K.M.M.); 3Centre for Metabolism, Obesity and Diabetes Research, McMaster University, Hamilton, ON L8S 4K1, Canada; 4Department of Medical Microbiology, School of Nutrition and Translational Research in Metabolism (NUTRIM), Maastricht University, 6229 ER Maastricht, The Netherlands; c.rossel@student.maastrichtuniversity.nl (C.A.J.R.); lienebervoets@gmail.com (L.B.); j.penders@maastrichtuniversity.nl (J.P.); 5Department of Obstetrics & Gynecology, McMaster University, Hamilton, ON L8S 4K1, Canada; simioni@mcmaster.ca (J.S.); li736@mcmaster.ca (J.L.); huttone@mcmaster.ca (E.K.H.); 6McMaster Midwifery Research Centre, McMaster University, Hamilton, ON L8S 4K1, Canada; 7Farncombe Family Digestive Health Research Institute, McMaster University, Hamilton, ON L8S 4K1, Canada; 8Department of Health Research Methods, Evidence and Impact, McMaster University, Hamilton, ON L8S 4K1, Canada; desouzrj@mcmaster.ca; 9Population Health Research Institute, Hamilton Health Sciences Corporation, Hamilton, ON L8L 2X2, Canada; 10Department of Epidemiology, Care and Public Health Research Institute (CAPHRI), Maastricht University, 6229 ER Maastricht, The Netherlands; monique.mommers@maastrichtuniversity.nl; 11InVivo Planetary Health: An Affiliate of the World Universities Network (WUN), West New York, NJ 10704, USA; 12Department of Medical Microbiology, Care and Public Health Research Institute (CAPHRI), Maastricht University Medical Centre, 6229 ER Maastricht, The Netherlands; 13Institute of Medical Microbiology, RWTH University Hospital Aachen, RWTH University, 52074 Aachen, Germany

**Keywords:** infant gut microbiome, microbial diversity, dietary diversity, 16S rRNA, infant nutrition, complementary foods, introduction to solids, gut community

## Abstract

The introduction of solid foods is an important dietary event during infancy that causes profound shifts in the gut microbial composition towards a more adult-like state. Infant gut bacterial dynamics, especially in relation to nutritional intake remain understudied. Over 2 weeks surrounding the time of solid food introduction, the day-to-day dynamics in the gut microbiomes of 24 healthy, full-term infants from the Baby, Food & Mi and LucKi-Gut cohort studies were investigated in relation to their dietary intake. Microbial richness (observed species) and diversity (Shannon index) increased over time and were positively associated with dietary diversity. Microbial community structure (Bray–Curtis dissimilarity) was determined predominantly by individual and age (days). The extent of change in community structure in the introductory period was negatively associated with daily dietary diversity. High daily dietary diversity stabilized the gut microbiome. Bifidobacterial taxa were positively associated, while taxa of the genus *Veillonella*, that may be the same species, were negatively associated with dietary diversity in both cohorts. This study furthers our understanding of the impact of solid food introduction on gut microbiome development in early life. Dietary diversity seems to have the greatest impact on the gut microbiome as solids are introduced.

## 1. Introduction

The gut microbiome refers to the bacterial ecosystem of the human gastrointestinal tract that consists of trillions of microbes [1] with a symbiotic relationship with the human host via metabolic, immunological, and nutritional functions [2]. Disturbances in the gut microbiome have been associated with numerous pathological states including obesity and atopy [3,4], which underlines the importance of a healthy gut microbiome. The infant gut microbiota mainly develops over the first 1–3 years of life, beginning as a relatively simple community with low richness and diversity, to one that resembles an adult-like state [5]. Important factors that influence early-life development of the gut microbiota are delivery mode [5,6], early feeding (breast milk versus formula) [5,7], exposure of infant and mother to antibiotics [8,9], probiotic usage [10], home environment after birth [11], gestational age at birth [12], geographical location, ethnicity [13], and the introduction of solid foods [6]. Diet impacts the taxonomy and function of microbial communities in the gut both in infancy [14] and in adulthood [15]. In infancy, the replacement of breastfeeding with formula impacts the relative abundance of early gut colonizers such as species of *Bacteroides* and *Bifidobacterium*, promotes specific shifts in bacterial metabolism and influences the rate of maturation of the gut microbiome overall [5,14]. 

The change from an exclusively milk-based diet to a solid food diet is a major event [16,17], however, we do not currently understand to what extent the choice of specific foods or the diversity of introduced foods influence microbial community diversity, structure, and taxonomy in the gut. The introduction of solid foods may initiate a shift towards an adult-like microbiota driven by changing ratios of fat, protein, carbohydrate, and fiber content in the diet [18]. Apart from studies examining breastfeeding versus formula feeding few studies have investigated early life nutritional exposures and the gut microbiota. Previous studies have shown that the introduction of solids results in changes in phyla abundances, especially an increase in Bacteroidetes [19,20]. There is no consensus on how microbial richness and diversity are affected, with individual studies showing an increase [19], no significant change [20] or a decrease after solid food introduction [21]. Of the reports that describe changes to the gut microbiota during the introductory period, none investigated the types of foods being introduced. In the current study, we evaluate the relationship between nutritional choices at the time of introduction to solid foods and gut bacterial dynamics in a cohort of full-term, vaginally born, and healthy infants from two geographically separated cohorts (Baby, Food & Mi, Canada and the LucKi-Gut study, The Netherlands). We used repeated sampling that allowed a high resolution (day-to-day) analysis of the changes occurring in the gut microbiome during this formative event.

## 2. Materials and Methods

### 2.1. Study Design

This study compares the effects of solid food introduction on the infant gut microbiome in infants living in Canada and the Netherlands. For the Canadian cohort, infants enrolled in the Baby & Mi study, a longitudinal cohort study carried out in Hamilton, Ontario [22], were asked to participate in the Baby, Food & Mi sub-study. For the Dutch cohort, infants were recruited to the LucKi-Gut study, a sub-study of the LucKi study, a longitudinal cohort study carried out in South-Limburg, the Netherlands [23]. Exclusion criteria for the present study at both sites were delivery via Caesarean section, admission to the neonatal intensive care unit, weaning from breast milk prior to the introduction of solids and receiving oral or IV antibiotics within four weeks of the beginning of the study. Although caregivers were advised not to wean infants from breast milk prior to the introduction of solid foods, one infant was exclusively formula fed at the time of solid food introduction. This infant was not excluded from the study due to the small sample size. Caregivers who provided consent for participation in the study collected daily stool samples from each infant starting approximately three days prior to the beginning of their planned introduction of solid food, then each day thereafter for up to two weeks. Caregivers provided food diaries for each day of the study period. In both cohorts, at least one sample was collected before the introduction of solids, and multiple samples were collected after the introduction of solids (Appendix A). The food diaries collected information about medications ingested during the study period. Infant use of antibiotics, maternal use of antibiotics, GBS prophylaxis and infant probiotic use were obtained from case report forms of the Baby & Mi study and the LucKi study. Data from additional stool samples from birth to one year of age (day 3, day 10, 6 weeks, 12 weeks, 5 months and 1 year) were also obtained from the Baby & Mi study and to 14 months from the LucKi study (1–2 weeks, 4 weeks, 8 weeks, 4 months, 5 months, 6 months, 9 months, 11 months, and 14 months). The Hamilton Integrated Research Ethics Board and Research Ethics Boards at other participating healthcare organizations approved the Baby, Food & Mi study. The LucKi-Gut study was approved by the Medical Ethics Committee Maastricht University Medical Center in the Netherlands.

### 2.2. Assessment of Nutritional Intake

Food diaries were harmonized prior to the start of data collection in both cohorts [24]. The collected food diaries were entered into the nutritional software Food Processor^®^ by ESHA© for the Baby, Food & Mi study and into the Dutch Nutrient Database (Nederlands Voedingsstoffenbestand—NEVO, RIVM, Bilthoven, The Netherlands) for the LucKi-Gut study. Caregivers reported the amount (in gram) or portion sizes in the food diaries and this information was used to estimate gram of each food. Calories from the macronutrients were calculated by the nutritional analysis software using the 4-4-9 rule, where each gram of protein or carbohydrate contributes 4 calories and each gram of fat contributes 9 calories to total caloric intake, with adjustment for fiber (2 kcal/g). Fiber (g/d) is an automatic output from the nutritional analysis software. Cumulative dietary diversity scores were calculated as follows: (1)$$\mathrm{food}\;\mathrm{diversity}\;\mathrm{score}=\;\frac{\mathrm{number}\;\mathrm{of}\;\mathrm{food}\;\mathrm{items}\;\times \;\mathrm{number}\;\mathrm{of}\;\mathrm{food}\;\mathrm{groups}}{\mathrm{number}\;\mathrm{of}\;\mathrm{days}}$$
(2)$$\text{pre-}/\mathrm{probiotic}\;\mathrm{diversity}\;\mathrm{score}=\;\frac{\left(2\;\times \;\mathrm{number}\;\mathrm{of}\;\mathrm{prebiotic}\;\mathrm{foods}+\mathrm{number}\;\mathrm{of}\;\mathrm{probiotic}\;\mathrm{foods}\right)}{\mathrm{number}\;\mathrm{of}\;\mathrm{days}}\;\times \;10.\;$$

Cumulative dietary diversity scores consider foods eaten over the entire study period. The food groups specified in this study were fruit, vegetables, grains (including beans and legumes), meat, dairy, confections/desserts, and oils. Beans and legumes were included in the grain category, to differentiate between meat and vegetarian protein. Foods considered prebiotic include garlic, onions, bananas, oats, apples, pears, flaxseed, wheat bran, whole grain, cruciferous vegetables, legumes, honey, coconut, berries, and corn products. Foods considered probiotic include anything fermented, e.g., yoghurt, pickled foods, cheese, tempeh, and sourdough. Daily dietary diversity scores were calculated based on the foods ingested in one day, using the same calculations for the cumulative dietary diversity scores, where the number of days = 1. The food diversity score was used instead of counting the number of food items and number of food groups, as it provides more information about the diet, including the distribution of food items across the food groups. Breast milk and formula were not included in the nutritional analysis.

### 2.3. Fecal Collection, DNA Extraction and 16S rRNA Gene Profiles

Fecal samples from 24 infants were collected from infants in both cohorts. Diaper liners (Bummis brand; Montreal, QC, Canada) were provided to participants in Canada, diaper liners were not used in the Netherlands. Participants were instructed to save the liner and/or stool within the collection bags provided. For fresh stool samples, only collected in Canada, the liner and stool were collected and processed within four hours of being produced. For frozen samples, the collection bag was stored in the home freezers until collected by research staff. Frozen samples were thawed on ice, and aliquots were made for metabolomic analysis then total genomic DNA was extracted from 0.1 g of stool with mechanical lysis with 2.8 mm ceramic beads and 0.1 mm glass beads for 3 min at 3000 rpm in 800 μL of 200 mM sodium phosphate monobasic (pH 8) and 100 μL guanidinium thiocyanate EDTA N-lauroylsarcosine buffer (50.8 mM guanidine thiocyanate, 100 mM ethylenediaminetetraacetic acid and 34 mM N-lauroylsarcosine), as previously described [25]. This extract was then purified with the MagMAX-96 DNA Multi-Sample Kit (Life Technologies, Carlsbad, CA, USA) on the MagMAX Express-96 Deep Well Magnetic Particle Processor (Applied Biosystems, Foster City, CA, USA). DNA was quantified using a Nanodrop 2000c Spectrophotometer (Thermo Scientific, Mississauga, ON Canada). The V3 16S rRNA gene was amplified for the Baby, Food & Mi samples according to the methods in our previous paper [26]. For the LucKi-Gut study, amplification of the bacterial 16S rRNA gene V3–V4 region was performed as previously described [26], with the following changes: the 319F and 806R primers were used, 5 pmol of primer, 200 μM of each dNTP, 1.5 mM MgCl_2_, 2 μL of 10 mg/mL bovine serum albumin, and 1.25 U Taq polymerase (Life Technologies, Carlsbad, CA, USA) were used in a 50 μL reaction volume. The PCR program used was as follows: 94 °C for 2 min followed by 30 cycles of 94 °C for 30 s, 50 °C for 30 s, and 72 °C for 30 s, then a final extension step at 72 °C for 10 min. Libraries were sequenced in the McMaster Genomics Facility with 2 × 250 bp reads on a MiSeq sequencer (Illumina, Inc.). Illumina sequences were demultiplexed with Illumina’s Casava software. Adapter, primer, and barcode sequences were trimmed from sequencing reads with cutadapt [27] and ASVs (amplicon sequence variants) were inferred with the Divisive Amplicon Denoising Algorithm 2 (DADA2) package [28] (1.16) in R. Taxonomy was assigned in DADA2 that uses the RDP naive Bayesian classifier method using the SILVA 16S rRNA gene reference file [29] (release version 132). In order to compare microbial community profiles between the studies, the raw sequences for the forward reads of the V3 sequences along with the V3 region of the forward reads from the V3–V4 sequences were processed together through DADA2 as above. Alpha diversity, the within community diversity, was estimated using observed richness and Shannon diversity that were calculated with the phyloseq package [30] (1.30.0) in R. Observed richness is an estimate of the number of species (in our case ASV) in each sample. Shannon diversity is calculated from the species richness and the evenness of the species distribution within each sample. Beta diversity, or the between sample diversity, was estimated using the Bray–Curtis dissimilarity, and analyzed with a principal coordinates analysis (PCoA), both calculated and plotted with the phyloseq package in R based on the relative abundance of ASVs. Permutational analysis of variance (PERMANOVA) and sample-to-sample changes in beta diversity were calculated with the vegan package [31] (2.5–7) in R. UPGMA trees were calculated using Bray–Curtis dissimilarity distance matrices of samples before and after the introduction of solid foods for merged V3 data from both cohorts with the stats package [32] (4.0.5). UPGMA trees were visualized using the ggtree package [33] (2.4.1). Phylogenetic trees were calculated by first aligning ASV sequences for each genus of interest (*Bifidobacterium*, *Veillonella* and *Bacteroides*) using DECIPHER [34] (2.18.1) then calculating pairwise distances and the neighbor joining tree and then optimizing the base frequencies and rate matrix with the phangorn package in R [35,36] (2.6.3) and visualized using the ape package [37] (5.4–1).

### 2.4. Statistical Analysis

Differences in the distribution of demographic data and nutritional variables between cohorts were assessed using unpaired *t*-tests, Welch tests and Chi-squared tests implanted through the GraphPad QuickCalcs Website [38]. For comparative analyses of alpha diversity before and after the introduction of solids, a Welch test was performed. Impacts of the different macronutrients, fiber and dietary diversity scores on alpha diversity were examined using linear mixed effects analyses using the lme4 package [39] (1.1-26) in R. The fixed effects varied by model and included calories from each of the macronutrients, fiber (g/d), dietary diversity, total calories from solid foods only, age in days, age at solid food introduction and use of intrapartum antibiotic prophylaxis, while a random effect for participant identifier (PID) is the same for all models. Fixed effects were included in the model if they were significantly associated with alpha diversity in univariate analyses. To analyze beta diversity, PERMANOVA with 9999 permutations was performed for each of the nutritional variables of interest. Generalized linear mixed effects models were used to analyze the association between sample-to-sample changes in beta-diversity and the nutritional variables of interest. The multivariable models were adjusted for estimated total energy from solid foods (kcal/d). The distribution used here was the gamma distribution, which was tested for using the fitdistrplus package [40] (1.1-3) and a Kolmogorov–Smirnov test. For analysis of specific bacterial taxa in relation to dietary intake, negative binomial regressions of bacterial ASV count data were used for the nutritional variables of interest using the glmmTMB package [41] (1.0.2.1). In the model, we added an offset for total counts to optimize the fit of the model to the data. Three models were fitted: (1) controlling only for PID, (2) controlling for PID and estimated energy intake from solid food (multivariate I) and (3) adjusting for estimated energy intake from solid food, PID and age at solid food introduction (weeks) (multivariate II). Bacterial taxa investigated in the negative binomial regressions were the ten most abundant ASVs for each cohort, as well as ASVs that were found to be associated with dietary data in DESeq analysis of ASV count data for the dietary variables of interest: food diversity (/d), pre-/probiotic diversity (/d) and fiber intake (g/d) (5 ASVs from each DESeq analysis with the greatest log2folddifference). DESeq was performed on the last sample of the study period for each infant only (cross-sectional). Heatmaps of the negative binomial regression results were constructed for simplified interpretation of the results. Spearman correlation matrices were generated for the metabolomics analyses. Corrections for multiple testing were not applied, due to the exploratory nature of this analysis. The threshold to declare statistical significance for all analyses presented here was *p* < 0.05.

## 3. Results

### 3.1. Study Population

Fecal samples were collected from 15 infants enrolled in the Baby, Food & Mi study and from 9 infants in the LucKi-Gut study. Baby, Food & Mi participants each provided between 3 and 6 (mean = 5) frozen and two fresh stool samples over the sub-study period (Appendix A) and LucKi-Gut participants each provided between 5 and 12 (mean = 9.4) frozen stool samples over the sub-study period (Appendix A). A total of 182 samples were collected in the Baby, Food & Mi study, and 89 samples in the LucKi-Gut study. The demographics of the two cohorts are shown in Table 1, the only significantly different variable was pre-pregnancy BMI (*p* = 0.043), which was higher in the Baby, Food & Mi study. All infants were full-term and vaginally born in both sub-studies. All infants were predominantly breastfed up until the time of solid food introduction in the Baby, Food & Mi study, although 20% of infants (*n* = 3) had some exposure to formula early in life (prior to 6 weeks). In the LucKi-Gut study, at the time of collection 7 infants were exclusively breastfed, 1 infant was exclusively formula fed and 1 infant was mixed fed. The median age at introduction of solids was 5.8 months (range, 4.0–6.5 months) in Canada and 5.0 months (range, 4.4–6.1 months) in the Netherlands. Only one infant (6.7%) and one mother (6.7%) were exposed to antibiotics (Abx) prior to the introduction of solid foods, while five infants (33.3%) received probiotics (Pbx) in Canada (2–17 weeks before solid food introduction). Neither infants nor mothers received antibiotics or probiotics before and throughout the study period in the Netherlands. Three mothers (20.0%) received intrapartum antibiotics (GBS prophylaxis) during delivery in Canada, while none of the mothers did in the Netherlands.

### 3.2. Nutritional Intake within Each Cohort at the Time of Solid Food Introduction

To assess dietary intake in the participant infants at the time of introduction of solids food diaries were collected for approximately two weeks in both cohorts (Canada: 8–15 days of solid food introduction; the Netherlands: 9–15 days of solid food introduction). There were marked differences in the types and number of food items introduced between the two cohorts. Infants in the Canadian sub-study were introduced to more food items and food groups than infants in the Dutch sub-study with a mean (SD) of 15 (7.3) food items in comparison to 8 (3.8) food items. None of the Dutch infants received meat, eggs, or dairy products (other than formula) at the time of solid food introduction, while 9 Canadian infants (64%) consumed meat and dairy products (meat and dairy products were always introduced together). These meat and dairy products included yoghurt, cheese, milk, cream cheese, animal meat, fish, and eggs. The Canadian infants were also introduced to a greater number of common food allergens including peanuts, wheat, eggs, fish, milk, and tree nuts (*n* = 11, 79%), while the Dutch infants were introduced to either wheat or celery (*n* = 4, 44%). An overview of the food items consumed by two or more infants in both cohorts can be seen in Figure 1, which demonstrates that most of the early food items were from the fruit and vegetables food groups. Commonly consumed food items in both cohorts include carrots, banana, and avocado. As previously mentioned, variation between the two cohorts is evident, as Canadian infants commonly consume more meat, dairy, and oil products, as well as baby cereals. In Canada, the majority of foods eaten by the infants, 66 food items, were only consumed by one infant, while 35 food items were consumed by two or more infants. In the Netherlands, infants showed more similar food item consumption, 12 food items were shared between two or more infants, while 16 were only consumed by one infant over the study period. Commercial baby foods and other processed foods were consumed very infrequently in both cohorts, with the exception of baby cereal, apple sauce and vegetable rice rusks in the Canadian cohort.

The estimated total energy, carbohydrate, fat, and protein intake from solids was higher in the Canadian sub-study than in the Dutch sub-study, whereas fiber intake (g/d) was similar between the two sub-studies (Table 2; Appendix A). These differences might be explained by the different solid food introduction approaches that appeared to be used. The majority of Canadian caregivers appeared to use a baby-led weaning approach, where infants receive more table foods [42,43], while the Dutch caregivers appeared to use a more traditional approach of beginning with small quantities of fruits and vegetables.

To analyze the variation in diet between the infants, dietary diversity scores were created. The food diversity score is based on the number of unique food items and number of food groups ingested, while the pre-/probiotic diversity score is calculated by adding the number of unique prebiotic and probiotic foods ingested by the infants, where prebiotic foods are weighted more heavily than probiotic foods. The cumulative food diversity score, the daily food diversity score and the daily pre-/probiotic diversity score were higher in Canada than in the Netherlands, and the range of scores was greater in the Canadian sub-study (Table 2). The general trend for the daily dietary diversity scores was an increase over time throughout the study period, despite many individual fluctuations, for most of the infants (Appendix A).

### 3.3. Microbial Community Structure Was Associated with Individual and Cohort

A merged analysis was undertaken to determine the effect of cohort in this study. Merged V3 data from the Baby, Food and Mi and the LucKi-Gut sub-studies was used for a multivariable PERMANOVA based on Bray–Curtis dissimilarities, which determined that individual infant was the strongest predictor of the gut microbial community (Figure 2A), despite the similar traits of the sub-study participants (e.g., born vaginally and breastfed prior to the introduction of solids). Cohort was the next strongest predictor of the gut microbiome (Figure 2A,B). Clustering of microbial profiles illustrates the inter-individual and cohort specific differences (Figure 2C,D) and showed that there is no common taxonomy change due to the introduction of solid food. Indeed, no association between study period (i.e., before or after solid food introduction) and gut microbial community structure was found in this merged analysis (Figure 2A). Taxonomic summaries highlight genus level similarities among infants from both cohorts with a high relative abundance of *Bifidobacterium*, *Bacteroides* and *Escherichia* at the time of solid food introduction. In fact, most of the infants in both studies were dominated by bifidobacteria (12/15 infants in Canada and 6/9 infants in the Netherlands). Infants without high relative abundance of bifidobacteria had high levels of *Bacteroides* in the Dutch cohort and either *Bacteroides* or Lachnospiraceae in the Canadian cohort. Genus level taxonomy was similar between sub-studies, 98.4% relative abundance of the genera in the Baby, Food and Mi study were shared with the LucKi-Gut study, while 99.7% relative abundance of the genera in the LucKi-Gut study were shared with the Baby, Food and Mi study. Differences between cohorts were apparent at the ASV level with only 222 of the 653 ASVs shared between the two cohorts (Figure 2E), indicating that while genera representation was similar between cohorts, there were species level differences. In view of the difference in ASV count between sub-studies, alpha diversity was investigated (Figure 2F). Observed richness was significantly higher in the Baby, Food & Mi sub-study than in the LucKi-Gut sub-study before (*p* < 0.05) and after the introduction of solids (*p* < 0.05), whereas Shannon diversity was not found to differ significantly. 

### 3.4. Alpha Diversity Increased over the Study Period

In light of the strong cohort effect, individual analysis of each sub-study was undertaken in the following sections. There was a significant increase in microbiome community richness and Shannon diversity after the introduction of solid food in the Canadian sub-study, after adjusting for individual infant (*p* < 0.05; Figure 3A,B). There was greater variation in Shannon diversity over the intensively sampled study period in the Canadian sub-study (0.37–3.10, $\bar{x}$_sub-study time point_ = 1.27–2.85, Figure 3B), compared with the Dutch sub-study, where the gut microbiota had more consistent Shannon diversity over time (0.59–2.82, $\bar{x}$_sub-study time point_ = 1.50–1.95, Figure 3H). Beyond this study period, out to approximately one year of age, alpha diversity increased over time and the one-year sample (or the 14-month sample in the Netherlands) had the highest values for both the Shannon index and observed richness (Figure 3C,D,I,J), which was significantly higher than the previous time points, with the exception of 11 months to 14 months in the Netherlands.

### 3.5. Alpha Diversity Was Associated with Dietary Variables at the Time of Solid Food Introduction

To assess the relationship between alpha diversity and the nutritional variables of interest, linear mixed effects models were performed. In the Canadian sub-study cumulative pre-/probiotic diversity score was positively associated with observed richness (β = 1.5, *p* < 0.05; Appendix A) and fiber was positively associated with Shannon diversity (Appendix A) after adjusting for estimated total energy from solid food (kcal/d), age (d), age at introduction and intrapartum antibiotic prophylaxis. In the Dutch cohort, fiber (g/d) (β = 3.3, *p* < 0.05), protein (kcal/d) (β = 1.6, *p* < 0.05), fat (kcal/d) (β = 0.27, *p* < 0.05) and daily food diversity scores (β = 0.64, *p* < 0.05) were positively associated with observed richness (Appendix A) but did not remain significant after adjustment for estimated total energy intake from solid food (kcal/d).

### 3.6. Increased Dietary Diversity Stabilized the Gut Microbiota at the Time of Solid Food Introduction

In the individual sub-study analyses, the introduction of solids had a significant impact on the gut microbial community in the Canadian cohort (*p* < 0.05, PERMANOVA, first and last sample for each infant, adjusted for individual infant and age at the time of introduction). The longitudinal nature of this study allowed for the measurement of the instability of the gut microbiome over time using the difference in Bray–Curtis dissimilarity from sample to sample over time within each infant (from day 3 to 1 year in Canada, from day 7 to 14 months in the Netherlands) and was found to be greater than 0.3 (median = 0.41–0.76) between study visits (Figure 3F,L). This underlines the unstable nature of the gut microbiome in early life. Beta diversity changed over the study period; however, each individual infant had its own trajectory, and no clear pattern emerged (Appendix A). Sample-to-sample changes in Bray–Curtis dissimilarity and the nutritional variables of interest were analyzed with generalized linear effects models and a Gamma distribution. After adjustment for estimated total caloric intake from solid food, protein intake (Netherlands), daily food diversity (Canada) and daily pre-/probiotic diversity (both sub-studies) were negatively associated with sample-to-sample changes in Bray–Curtis dissimilarity (Table 3), implying that protein and higher daily dietary diversity stabilized the gut microbiota during the period of solid food introduction.

### 3.7. Bacterial ASVs Associated with Nutritional Variables

As alpha and/or beta diversity were found to be associated with fiber (g/d), food diversity score (/d) and pre-/probiotic diversity score (/d), we next examined which bacterial ASVs were associated with these nutritional variables by DESeq analysis on the last sample of the study period. In Canada, fiber intake was negatively associated with five *Bacteroides* ASVs, one *Clostridium* sensu stricto 1 ASV, one *Shimwellia* ASV, one *Escherichia/Shigella* ASV, one *Bifidobacterium* ASV, one member of the Enterobacteriaceae family and one member of the Atopobiaceae family. Daily food diversity score was negatively associated with ASVs from the genera *Veillonella, Bacteroides, Citrobacter, Parabacteroides* and *Phascolarctobacterium*. Daily pre-/probiotic diversity scores were positively associated with members of the genera *Veillonella, Bacteroides, Bifidobacterium, Escherichia/Shigella, Parabacteroides* and the Atopobiacaeae family (Appendix A). In the Netherlands, fiber intake was negatively associated with three *Clostridium* ASVs, one *Bifidobacterium* ASV and one *Proteus* ASV. Food diversity score (/d) was negatively associated with three *Bacteroides* ASVs, one *Escherichia* ASV and one *Sutterella* ASV (Appendix A). The association of individual ASVs with dietary variables while accounting for repeated sampling and controlling for confounding variables was examined next. In order to narrow the scope of analysis, five ASVs with the highest log_2_fold difference per nutritional variable and the 10 most abundant ASVs per sub-study were included in negative binomial regression analyses. 

Despite the differences in bacterial communities and dietary intake between the Canadian and Dutch sub-studies, diet diversity was positively associated with members of the genus *Bifidobacterium* and negatively associated with members of the *Veillonella* genus in both sub-studies (Figure 4). Dietary diversity was also positively associated with an Enterobacteriaceae ASV and two members of the *Bacteroides* (Canada) and negatively associated with a different *Bacteroides* ASV (Canada) and an ASV for *Clostridium neonatale* (Netherlands). Daily pre-/probiotic diversity was positively associated with a member of the genus *Veillonella* in Canada (Figure 4A). Fiber intake was negatively associated with a different *C. neonatale* ASV and positively associated with a *Proteus* ASV, and protein intake was positively associated with one *Bifidobacterium* ASV and negatively associated with another in the Dutch sub-study (Figure 4B). Fat in the diet was negatively associated with a *Veillonella* ASV and a *Bacteroides* ASV that was also positively associated with carbohydrates in the diet, in the Canadian sub-study (Figure 4A). These findings suggest species-specific responses to the infant diet during the first weeks on solid food. Results for the unadjusted models, and the models adjusted for caloric intake were investigated and demonstrated some of the same associations (Appendix A). 

Due to the similarity in the response of members of the genera *Bifidobacterium, Bacteroides* or *Veillonella* to dietary variables across the sub-studies, which was a robust finding, the possibility that the same bacterial species were being represented in these two geographically distinct sub-studies was investigated. Phylogenetic trees for each genus, constructed from ASV sequences, showed a similarity between *Veillonella* ASVs that were lower with increasing food diversity (Appendix A), suggesting a similar species response to the diet. No other clustering of ASVs that had the same association with the diet variables between sub-studies was found (Appendix A). This suggests overall similarities in the bacterial response to diet at the genus level across sub-studies but not at the species level.

## 4. Discussion

Despite being one of the most important events during infancy [16,17], the association between the introduction of solid foods and the infant gut microbiome remains understudied. In fact, previous studies have solely looked at general changes in the gut microbiome after solid food introduction without investigating specific food choices and macronutrient composition of the diet. In this study, we characterized the infant gut microbiomes of 15 Canadian and 9 Dutch healthy, full-term infants. We related the gut microbiome to each infant’s daily nutritional intake from solid foods for two weeks during solid food introduction to further our understanding of the impact of nutritional exposures on the gut microbiota in early life. The day-to-day sampling allowed for a high-resolution analysis of the infant gut microbiome during this dietary event. As expected, the gut microbiome of infants at the time of solid food introduction was highly dependent on the individual [5], but we describe the novel findings that dietary diversity was associated with microbiome stability throughout the introductory period and that the bacterial genera *Bifidobacterium*, *Bacteroides* and *Veillonella*, had similar responses to the diet in both cohorts, indicating that there are diet-specific responses to the introduction of solid foods despite differences in geographical location. 

In this study, the gut microbiome of infants at the time of solid foods showed high inter-individual variability. There were also strong cohort effects, which was expected, as the composition of the gut microbiome is known to vary by geographical area [13]. We saw a decrease in inter-individual variability as age increased beyond the introductory period, which was in accordance with the literature [5]. Consistent with previous studies, alpha diversity increased over time from the first days after birth to one-year of age (14 months in the Netherlands) [5,44]. In the 2-week intensively studied period, Shannon diversity and observed richness increased significantly in the Canadian cohort, but were not found to increase significantly in the Dutch population, which could be due to the different approaches to solid food introduction in the two cohorts, with Canadian infants being introduced to an average of 15 food items from all the food groups and common allergen categories and the Dutch infants being introduced to an average of 8 food items that did not include meat, eggs, dairy or common food allergens except wheat. In Canadian infants, fiber was positively associated with increased alpha diversity. In a study performed in mice, a diet with low fiber led to decreased levels of alpha diversity [45], while other studies found that a diet high in fruit, vegetables and fiber resulted in higher bacterial richness and diversity in adult humans [2,46]. Our findings in infants, together with these studies in animals and human adults, suggest that the effect of fiber on the infant gut may be consistent across the life course. 

Most infants in this study had a gut microbiota dominated by bifidobacteria or *Bacteroides* at the time of solid food introduction. This was expected, as the majority of infants were exclusively breastfed and HMOs in breast milk support the growth of bifidobacteria and lactic acid bacteria, which can utilize metabolized HMO components, leading to a high relative abundance of Actinobacteria and Firmicutes [47,48]. A previous study reported an increase in *Bacteroides, Ruminococcus* and a decrease in *Escherichia* abundance in the introductory period, however it was not specific to the kinds of food ingested [20]. Carbohydrate intake in the Canadian cohort was positively associated with the relative abundance of *Bacteroides* and protein intake was positively associated with a number of bifidobacterial ASVs in the LucKi-Gut study.

One of the main findings of this study was that dietary diversity was positively associated with stability of the microbial community, e.g., less change during the introductory period, in both Canadian and Dutch infants suggesting that a diverse diet stabilizes the composition of the gut microbiota during solid food introduction. Alpha diversity was positively associated with dietary diversity in Canadian infants. Increased dietary diversity has been correlated with alpha diversity [49,50] and stability of the gut microbiome in adults [51] and may be due to increased functional capacity of the microbial community in the gut when exposed to a variety of substrates. Increased microbial diversity is believed to stabilize the gut microbiome, due to increased functional redundancy allowing for the gut microbiota to withstand perturbations [52,53]. Thus, introducing a high variety of first foods may increase alpha diversity and stabilize the gut microbiome early in life, however, the long-term implications of these associations remain unknown.

Although dietary intake was markedly different between cohorts, we found similarities in the response of some bacterial groups to the same dietary variables in both cohorts. A member of the genus *Veillonella*, that we suspect is the same species, was negatively associated with cumulative dietary diversity and with pre-/probiotic diversity in both cohorts. The abundance of *Veillonella* has previously been positively associated with continued breastfeeding and has been named a protective factor in early life [54]. The negative association of *Veillonella* with dietary diversity might be explained by an increase in number and amount (total kcal) of foods, decreasing the intake of breast milk over the sub-study period. We also found that members of the genus *Bifidobacterium* were positive associated with dietary diversity in both cohorts, although they did not appear to be from the same species. These findings indicate that dietary diversity is beneficial to a healthy gut microbiome, as bifidobacteria have many beneficial effects on the human host, for example competition with pathogenic bacteria, and are, therefore, desired in the gut community [55]. 

To our knowledge, this study was the first to collect detailed dietary data on a day-to-day basis during an important dietary milestone in infancy followed by evaluation of the gut microbiome to 12–14 months of age. This allowed for a high-resolution analysis of the changes occurring in the gut microbiome during solid food introduction. Overall, this study established that the introduction of solids induces gradual changes in the infant gut microbiome, rather than rapid and stark differences. Since the majority of infants are still receiving breast milk at the time of solid food introduction, more dramatic changes might occur if breast milk constituents, such as HMOs, are eliminated as a substrate for the gut microbiota. The variety of foods introduced, as well as their macronutrient composition influences the changes occurring in the gut microbiome, which was not investigated previously. It also provided a detailed insight into the approach in which solid foods are introduced in a healthy population in different countries. The introduction of solids is one factor impacting the development of the gut microbiome during infancy. Interestingly, the infant gut microbiome behaves similarly to the adult gut microbiome in terms of dietary diversity and its association with alpha diversity and the stability of the gut community. Early-life influences may have long-term health implications, as the introduction of solid foods starts the trajectory toward the diversity and composition of the gut microbiota in the adult.

Strengths of our study include the homogeneity of the study population within each cohort that allowed for significant associations to be detected at each study location; the repeated longitudinal sampling that allowed the observation of day-to-day changes in the diet and gut microbiome; the detailed measurements of dietary data; and a standard laboratory and analysis protocol that was applied to samples from both locations. Limitations include the small study populations that limited statistical power and generalizability; the uncertainty in total calorie calculations due to the exclusion of breastfeeding and the inherent estimation of portion sizes with infant feeding; the potential masking of the effects of solid food nutrition due to continued feeding with breast milk that has been seen in other studies [16]; and the limited ability to assign bacterial species due to the taxonomic resolution of 16S rRNA gene amplicons.

## 5. Conclusions

This study demonstrates that the introduction of solid foods has an impact on the developing infant gut microbiome and that nutritional choices influence the changes that occur, however, cohort and individual differences were the most prominent factors determining differences in the gut microbiota. Higher fiber intake and high dietary diversity were associated with higher alpha diversity. High daily dietary diversity was associated with stability of the gut microbiota over the study period, as seen previously in adults [51]. This underlines the importance of a healthy, fiber-rich, and diverse diet throughout the life-course. Further research is needed to understand the whole ecosystem of the infant gut microbiome and to describe bacterial responses to dietary changes in infancy. Overall, this study contributes new knowledge to the research topic of the development of the gut microbiota in infancy and the influences of early dietary choices. 

## Figures and Tables

**Figure 1 nutrients-13-02639-f001:**
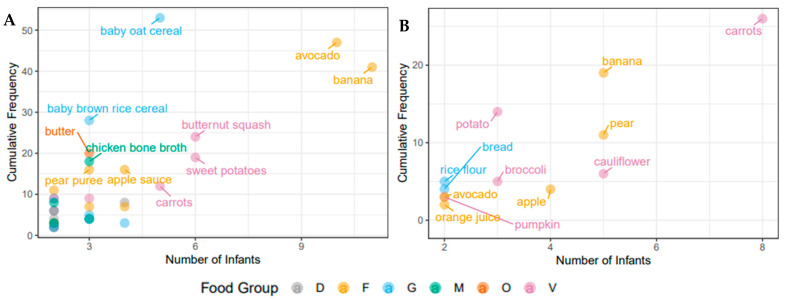
Individual food items consumed by the infants over the study period. (**A**) Baby, Food & Mi sub-study; (**B**) LucKi-Gut study. Only food items that were eaten by two or more infants are shown. Cumulative frequency describes overall intake of servings over the study period. The food groups shown here are dairy (D), fruit (F), grains (G), meat (M), oils (O) and vegetables (V).

**Figure 2 nutrients-13-02639-f002:**
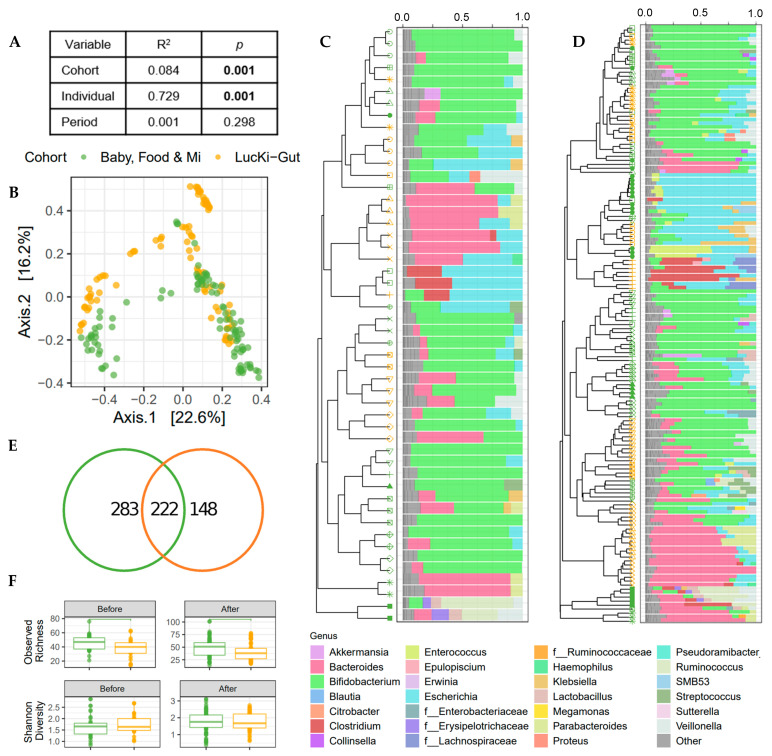
Bacterial communities differ by cohort; however, greater variability is seen between individuals. (**A**) Explained variance (R^2^) and statistical significance of cohort (Baby, Food & Mi/LucKi-Gut sub-studies), individual and period (before/after solid food introduction) on the microbial community structure (Bray–Curtis dissimilarity) as determined by multivariable PERMANOVA on merged V3 data. (**B**) PCoA of microbial communities. (**C**,**D**) Dendrograms of UPGMA clustering of individuals by Bray–Curtis dissimilarity and relative genus abundance bar charts for each sample before the introduction of solids (**C**) and after the introduction of solids (**D**). Color of symbols indicate sub-study and shape of symbols indicate individual. Bacterial genera in “Other” include genera with a relative abundance < 5%. (**E**) Venn diagrams of the number of ASVs shared between the sub-studies. (**F**) Boxplots depicting the observed richness and Shannon diversity before and after the introduction of solid foods. For all panels green = Baby, Food & Mi, orange = LucKi-Gut.

**Figure 3 nutrients-13-02639-f003:**
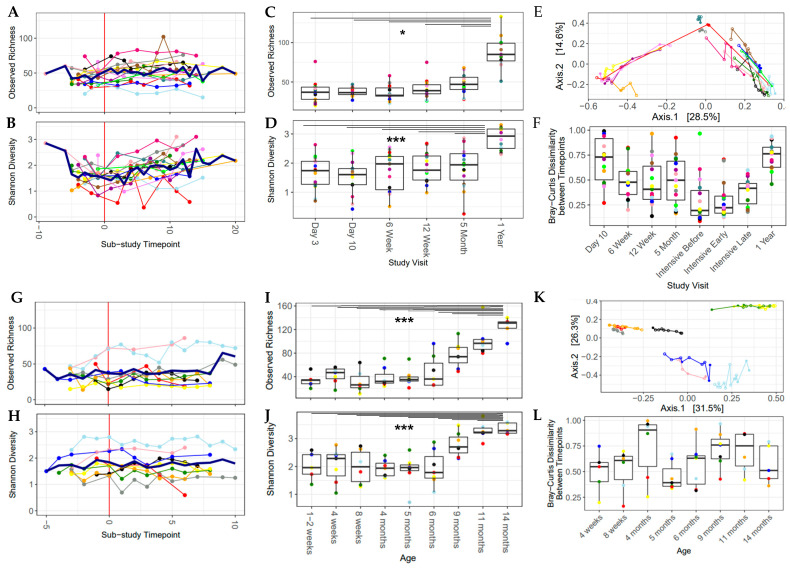
Alpha diversity increased over the sub-study period and over the first year of life. Bray–Curtis dissimilarity over time shows that the gut microbial community was dynamic over the first year of life. All plots are colored by participant ID (PID). (**A**–**F**) Baby, Food & Mi sub-study; (**G**–**L**) LucKi-Gut sub-study. (**A**,**G**) Observed richness over the sub-study period, mean values are shown with a thick blue line. (**B**,**H**) Shannon diversity over the sub-study period, mean values are shown with a thick blue line. (**C**,**I**) Observed richness over the first year of life. (**D**,**J**) Shannon diversity over the first year of life. (**E**,**K**) PCoA plot of the sub-study samples. (**F**,**L**) Change in Bray–Curtis dissimilarity between each study visit and the one previous to it. (* *p* < 0.05, *** *p* < 0.001).

**Figure 4 nutrients-13-02639-f004:**
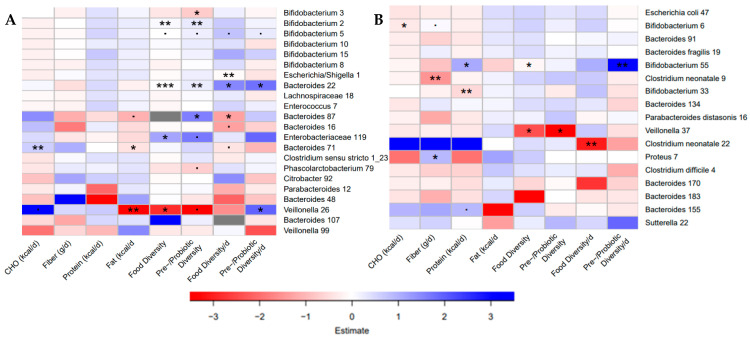
Bacterial taxa in both communities were associated with dietary variables. (**A**) Heatmaps of the associations found in negative binomial regressions when adjusted for total caloric intake (kcal/d) and age at introduction (weeks) in the Baby, Food & Mi study. (**B**) Heatmaps of the associations found in negative binomial regressions when adjusted for total caloric intake (kcal/d) and age at introduction (weeks) in the LucKi-Gut study. Negative associations are shown in red, while positive associations are shown in blue. (*p* < 0.1, * *p* < 0.05, ** *p* < 0.01, *** *p* < 0.001).

**Table 1 nutrients-13-02639-t001:** Demographics of the study population.

	Baby, Food & Mi				LucKi-Gut				
	*n* (%)	Median	Range	Mean (SD)	*n* (%)	Median	Range	Mean (SD)	Sig. ^1^
Age at Introduction (months) ^++^	15 (100)	5.79	3.98, 6.5	5.5 (0.66)	9 (100)	5.03	4.37, 6.12	5.2 (0.66)	
Gestational Age at Birth ^2,+^									
Completed 37 weeks	0 (0)	-	-	-	1 (14)	-	-	-	
Completed 38 weeks	3 (20)	-	-	-	0 (0)	-	-	-	
Completed 39 weeks	4 (27)	-	-	-	1 (14)	-	-	-	
Completed 40 weeks	6 (40)	-	-	-	2 (29)	-	-	-	
Completed 41 weeks	2 (13)	-	-	-	2 (29)	-	-	-	
Completed 42 weeks	0 (0)				1 (14)	-	-	-	
Female Sex ^+^	7 (47)	-	-	-	4 (44)	-	-	-	
Parity ^2,+^									
0	6 (40)	-	-	-	4 (57)	-	-	-	
1	4 (27)	-	-	-	2 (29)	-	-	-	
2	5 (33)	-	-	-	1 (13)	-	-	-	
Hospital Birth (Y) ^3,+^	9 (60)	-	-	-	5 (71)	-	-	-	
Gestational Diabetes (Y)	4 (27)	-	-	-	-	-	-	-	
Pre-pregnancy BMI (kg/m^2^) ^++^	15 (100)	23.5	17.7, 24.0	24.1 (4.16)	7 (78)	19.9	18.6, 24.7	20.5 (1.97)	*
GBS Prophylaxis (Y) ^+^	3 (20)	-	-	-	0 (0)	-	-	-	
Infant Oral Abx (Y) ^+^	1 (7)	-	-	-	0 (0)	-	-	-	
Infant Oral Pbx (Y) ^+^	5 (33)	-	-	-	0 (0)	-	-	-	
Maternal Oral Abx (Y) ^+^	1 (7)	-	-	-	0 (0)	-	-	-	

^1^ *: *p* < 0.05. ^++^ Differences between cohorts were tested using unpaired *t*-tests for continuous variables. ^+^ Differences between cohorts were tested using Chi-squared tests for categorical variables. ^2^ Collected for 7/9 infants in the LucKi-Gut study. ^3^ Hospital birth means infants were born in hospital; the alternative is home birth. Collected for 7/9 infants in the LucKi-Gut study. Y = Yes.

**Table 2 nutrients-13-02639-t002:** Overview and comparison of the average nutritional intake over the study period.

	Baby, Food & Mi		LucKi-Gut		
	Mean (SD)	Range (Min, Max)	Mean (SD)	Range (Min, Max)	Sig. ^1^
Estimated total energy from solid food (kcal/d)	53.9 (64.06)	468.8 (0.00, 468.8)	28.9 (24.5)	109 (0.00, 109.00)	***
Carbohydrates (kcal/d)	25.7 (32.45)	167.9 (0.00, 167.9)	18.5 (19.9)	94.8 (0.00, 94.8)	*****
Fat (kcal/d)	21.2 (34.71)	246.9 (0.00, 346.9)	6.6 (11.3)	70.2 (0.00, 70.2)	***
Protein (kcal/d)	6.92 (9.07)	60.7 (0.00, 60.7)	1.9 (2.1)	11.2 (0.00, 11.2)	***
Fiber (g/d)	1.0 (1.62)	8.98 (0.00, 8.98)	0.8 (0.8)	3.90 (0.00, 3.90)	
Food Diversity	6.34 (5.05)	14.8 (0.800, 15.6)	2.9 (2.8)	8.90 (0.40, 9.30)	*
Pre-/Probiotic Diversity	7.8 (5.54)	16.7 (0.00, 16.7)	5.9 (2.5)	10.9 (2.00, 12.9)	
Food Diversity (/d)	8.31 (9.80)	55.0 (0.00, 55.0)	4.2 (4.7)	20.0 (0.00, 20.0)	***
Pre-/Probiotic Diversity (/d)	25.1 (24.15)	100.0 (0.00, 100.0)	15.2 (15.2)	60.0 (0.00, 60.0)	***

^1^ * *p* < 0.05, *** *p* < 0.001. Differences between cohorts were tested using Welch tests for continuous variables. Abbreviations: d, day.

**Table 3 nutrients-13-02639-t003:** Results of the generalized linear mixed effects model for sample-to-sample within-subject changes in microbial community structure (Bray–Curtis dissimilarity index) and the demographic/nutritional variables of interest for both cohorts.

	Baby, Food & Mi				LucKi-Gut			
	Univariable		Multivariable ^1^		Univariable		Multivariable ^1^	
	Estimate	Sig.	Estimate	Sig.	Estimate	Sig.	Estimate	Sig.
Age (d)	−0.0010	0.80	-	-	−0.0052	0.20	-	-
Sex	−0.17	0.28	-	-	−0.16	0.31	-	-
Total energy (kcal/d)	−0.0028	**0.025**	-	-	−0.0001	0.97	-	-
Carbohydrates (kcal/d)	−0.0043	0.12	−0.0029	0.15	−0.0004	0.91	−0.0016	0.80
Fiber (g/d)	−0.069	0.17	0.043	0.60	−0.025	0.76	−0.048	0.68
Protein (kcal/d)	−0.019	**0.036**	−0.0077	0.68	−0.045	0.12	−0.11	**0.01**
Fat (kcal/d)	−0.044	**0.030**	0.0019	0.33	0.0028	0.58	0.0033	0.60
Food Diversity (/d)	−0.033	**0.0003**	−0.026	**0.0079**	−0.008	0.62	−0.0088	0.60
Pre-/Probiotic Diversity (/d)	−0.016	**<0.0001**	−0.014	**0.00014**	−0.0077	**0.007**	−0.008	**0.039**

^1^ Multivariable models are adjusted for total caloric intake from solid foods (kcal/d). Significant associations are represented in bold.

## Data Availability

The data presented in this study are available on request from the corresponding authors (J.C.S., N.v.B.). The data are not publicly available due to the potentially identifiable nature of the data and privacy concerns by study participants.

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
