# Peer review of "Infants’ First Solid Foods: Impact on Gut Microbiota Development in Two Intercontinental Cohorts"

_nutrients, 2021, doi:10.3390/nu13082639_

Round 1

Reviewer 1 Report

In the abstract, consider changing 'solids' to 'solid foods' in the first sentence, especially as this is the first time this term is introduced.

In Fig. 1, perhaps 'other' could come at the end, after 'veillonella'? It's presently placed in alphabetical order between 'megamonas' and 'parabacteroides'.  If too much effort to make this change, can keep as-is.

Although 'specific food choices' is mentioned in the first paragraph of the discussion, the tables and figures only feature macronutrients (e.g., carbohydrate, protein) and fiber (a carbohydrate).  It would have been interesting to have featured the actual foods consumed.  With such small sample sizes in both of the two sites and only 2 weeks of intensive data collection, would this have been feasible? Because dietary diversity was addressed, presumably based on consumption of a variety of foods from specific categories, the specific foods consumed (and their quantity) seem to have been known.

In Figure S6 (in Supplementary Materials), fiber is spelled 'fibre' -- check for consistency of spelling throughout.

Lines 158-159:  "The V3 16S rRNA 158
gene was amplified for the Baby, Food & Mi samples according to [26]." This sentence should be completed, with the reference at the end.

Line 316:  The first time 'Shannon diversity' is introduced, it would be helpful if a one-sentence definition or explanation accompanies this. Not all readers (including this reviewer) are familiar with Shannon diversity.  Related to this, alpha diversity is introduced in Line 175.  How is alpha diversity different to Shannon diversity?

header 3.4 (Line 329) refers to alpha diversity, but nowhere in the paragraph is alpha diversity addressed -- only Shannon diversity. 

header 3.5 (Line 350) also refers to alpha diversity. The first sentence refers to alpha diversity but later Shannon diversity is mentioned.  I don't understand the difference between alpha and Shannon diversity.

Bray-Curtis dissimilarity index is featured throughout the manuscript, but it isn't ever explained or defined.  What does it mean and how it is relevant to the points being made?

Lines 520-522: "In- 520
terestingly, the infant gut microbiome behaves similarly to the adult gut microbiome in 521
terms of dietary diversity and its association with alpha diversity and the stability of the 522
gut community."  How is stability defined and why is it important? Is a dynamic gut microbiome of concern or problematic?

Reviewer 2 Report

Thank you for this interesting paper. I agree that the role of introduction of solid food on microbiome structure and function has not been sufficiently studied.

The authors included a small number of infants but the diet data is extensive and the information important.

I have only a few questions:

1) Did you have any information on commercial vs. home-made baby food as that could affect the outcomes

2) Why did you not just simply counted number of foods and number of food groups as two different scores

3) What was superior in terms of microbial diversity – food diversity or pre/probiotic diversity

4) There was a large number of children born in Canada by c-section and used antibiotics – did that affect your outcomes

5) I thought you said in the methods that you only included BF infants, but you showed results of one infant that was FF

6) Do you have data to show how increase in intake of each additional food may affect diversity?

7) Can you prepare a figure of single foods vs. microbial diversity as you have done in figure 3 for single bacteria i.e. are certain foods or food groups driving gut microbiome diversity more than others?
